# Timbral cues underlie instrument-specific absolute pitch in expert oboists

**Niels Chr. Hansen** [ID][1,2,3,4]☉ *, **Lindsey Reymore**[5,6]☉

**1** Aarhus Institute of Advanced Studies, Aarhus University, Aarhus, Denmark, **2** Centre of Excellence in Music, Mind, Body, & Brain, Department of Music, Art and Culture Studies, University of Jyväskylä, Jyväskylä, Finland, **3** Interacting Minds Centre, School of Culture and Society, Aarhus University, Aarhus, Denmark, **4** Royal Academy of Music, Aarhus/Aalborg, Denmark, **5** Schulich School of Music, McGill University, Montreal, Canada, **6** School of Music, Dance, and Theatre, Arizona State University, Tempe, AZ, United States of America

☉ These authors contributed equally to this work.
* nchansen@aias.au.dk

**Data Availability Statement:** All stimuli, analysis scripts, and anonymized data files are available

## Abstract

While absolute pitch (AP)—the ability to identify musical pitches without external reference—is rare even in professional musicians, anecdotal evidence and case-report data suggest that some musicians without traditional AP can nonetheless better name notes played on their musical instrument of expertise than notes played on instruments less familiar to them. We have called this gain in AP ability "instrument-specific absolute pitch" (ISAP). Here, we report the results of the first two experiments designed to investigate ISAP in professional oboists. In Experiment 1 ($n = 40$), superiority for identifying the pitch of oboe over piano tones varied along a continuum, with 37.5% of oboists demonstrating significant ISAP. Variance in accuracy across pitches was higher among ISAP-possessors than ISAP-non-possessors, suggestive of internalized timbral idiosyncrasies, and the use of timbral cues was the second-most commonly reported task strategy. For both timbres, both groups performed more accurately for pitches associated with white than black piano keys. In Experiment 2 ($n = 12$), oboists with ISAP were less accurate in pitch identification when oboe tones were artificially pitch-shifted. The use of timbral idiosyncrasies thus may constitute a widespread mechanism of ISAP. Motor interference, conversely, did not significantly reduce accuracy. This study offers the first evidence of ISAP among highly trained musicians and that reliance on subtle timbral (or intonational) idiosyncrasies may constitute an underlying mechanism of this ability in expert oboists. This provides a path forward for future studies extending the scientific understanding of ISAP to other instrument types, expertise levels, and musical contexts. More generally, this may deepen knowledge of specialized expertise, representing a range of implicit abilities that are not addressed directly in training, but which may develop through practice of a related skill set.

from the Open Science Framework (OSF) (https://doi.org/10.17605/OSF.IO/HS2M9).

**Funding:** While conducting this work, NCH received funding from the European Union's Horizon 2020 research and innovation programme under the Marie Skłodowska-Curie grant agreement No 754513, the European Research Council (ERC) under the European Union's Horizon Europe research and innovation programme grant agreement No 101045747, and The Aarhus University Research Foundation. At the start of the project, LR was a postdoctoral fellow with the ACTOR Project (Analysis, Creation, and Teaching of ORchestration), funded by the Social Sciences and Humanities Research Council (SSHRC) and McGill University. The ACTOR Project also provided funding for this study. Views and opinions expressed are those of the authors only and do not necessarily reflect those of the European Union or European Research Council Executive Agency. Neither the European Union nor the granting authority can be held responsible for them. The funders had no role in study design, data collection and analysis, decision to publish, or preparation of the manuscript.

**Competing interests:** The authors have declared that no competing interests exist.

# Introduction

Expertise constitutes an important topic of study germane to fields such as psychology, cognitive science, and artificial intelligence, where the effects associated with intense, long-term training have been investigated in relation to physiological adaptations and complex cognitive mechanisms [1]. While expertise is often studied in relation to explicit decision-making, as in chess, sports, and military strategy [2], expert performance nearly always entails implicit competences as well [3]. Music is especially informative in studying implicit skill acquisition because of the large variation in experience levels among the general population and because professional musicians undergo intensive periods of training [4] associated with notable implicit stylistic enculturation [5].

Absolute pitch (AP), or the ability to identify musical pitches without external reference to another known pitch, is thought to arise from an interaction of innate and experiential factors [6]. This ability is generally difficult to acquire through explicit training, particularly after an apparent critical period during childhood (however, see [7]) and is rare even in professional musicians [8, 9]. Some AP possessors may demonstrate variance in the strength of their abilities in relation to the harmonic complexity of a note and/or its timbre (e.g. [10, 11]). Nevertheless, AP is often associated with individuals who are able to identify pitches with high accuracy across a range of timbres; we refer to this type of AP as "global" AP. Yet, anecdotal evidence and case-report data suggest that some expert musicians who do not possess "global" absolute pitch as such are nonetheless better able to name notes played on their primary musical instrument of expertise than notes played on other instruments that are less familiar to them. Such an ability would suggest that intensive long-term training may lead to the development of a specific variety of absolute pitch ability in at least some musicians.

In a previous paper [12], we referred to this gain in absolute pitch identification ability for one's own instrument type as "instrument-specific absolute pitch" (ISAP) and proposed a theory of the potential underlying mechanisms, which we suggest are developed implicitly in at least some musicians during long-term training. Those musicians in our theorized category of "ISAP possessors" would demonstrate a significantly higher pitch identification accuracy for notes played on their primary instrument of expertise as compared to notes played on other instruments. That is, we would expect oboists with ISAP to be able to identify oboe tones more accurately than flute or piano tones, and we would expect flautists with ISAP to be able to identify flute tones more accurately than oboe or piano tones. We have proposed that instrument-specific absolute pitch, which we theorize develops from expert-level familiarity with a musical instrument's timbre, may use mechanisms distinct from global absolute pitch as it has traditionally been considered.

It should be noted that we do not have reason to believe that global AP and ISAP are mutually exclusive; a given individual may display characteristics of neither ability, one, or both. Specifically, we have operationalized ISAP as the extent to which pitch-labeling performance for one's primary instrument exceeds performance for other, less familiar instruments—the difference in performance between instruments defines ISAP. Consider three subgroups of people with varying levels of global AP: the first scores at chance level across timbres ("non-AP"), another subgroup scores above chance but below ~90% across timbres ("quasi-AP"), and yet another subgroup scores above ~90% across timbres ("AP"; for discussion of "quasi" or "partial" AP, see [13, 14]). Members of each of these subgroups could have an added advantage for the timbre of one or more of their main instruments, and this added advantage is what we would refer to as "ISAP"—ISAP is thus an instrument-specific *gain* in absolute pitch ability. Note, however, that ISAP would offer little additional advantage and thus would be nearly impossible to detect in global AP possessors whose overall accuracy is near to 100%, as there would be a ceiling effect.

Only a few previous studies have tested the possibility that there may be a pitch-naming advantage for one's primary instrument in individuals who do not possess strong global AP. In violinists and pianists without global AP, Wong and Wong [15] found that instrumentalists were better able to identify pitches played on their own instruments as compared to sine tones; however, instrumentalists were not tested on non-primary instruments for comparison, and thus the difference in accuracy could plausibly be related to the harmonic complexity of the tones, rather than the familiarity of the timbre (cf. [16, 17]). Marvin and Brinkman [18] asked violinists and pianists without global AP to identify synthesized violin and piano tones. While pianists showed an advantage for piano over violin tones, the overall performance of the violinists was not significantly different between the two timbres. Although the authors did not make this proposal, one could imagine this possible difference in ISAP between violinists and pianists could be due to the fact that general aural skills are often practiced with piano timbre, and many non-pianists are expected to acquire basic piano skills. In a group of 12 musicians without global AP, Schlemmer, Kulke, Kuchinke, and Van Der Meer [19] only found a pitch-naming advantage for white key notes played with a familiar timbre of a musical instrument that the participants had taken lessons in.

Finally, Li [20] did not observe a pitch identification advantage for string timbres in string majors; however, while Li's participants were not asked to self-identify as either having or not having global AP, they were mostly musicians with very high degrees of global AP ability (as determined in a previous experiment) who began musical training early. We propose that global AP typically recruits a different set of mechanisms for absolute pitch identification than ISAP does. Thus, we do not expect that most global AP possessors will demonstrate use of the mechanisms we propose for ISAP, and accordingly, we would not necessarily expect a similar level of primary-over-other-instrument advantage from these individuals.

Each of these studies has pursued a group-level advantage for primary instrument timbre; however, based on anecdotal evidence and case-report results [12], we suggest that as with global AP, it is likely the case that not all musicians have ISAP or may at least have different degrees of ISAP. In this scenario, research would need to first identify individuals with ISAP. This approach contrasts with the conventional group-level approach of testing for the ability, which assumes that most or all musicians will exhibit an advantage for their primary instrument timbre.

The first proposed mechanism of our theory of ISAP [12] entails that musicians with ISAP use timbral cues specific to the type of instrument that they play (oboe, clarinet, etc.) to aid in pitch identification. Such cues may be available via two routes. On one hand, evidence from neuroimaging studies suggests that extreme familiarity with one's primary instrument has a marked impact on auditory processing [21–23]. Thus, increased or better coordinated cortical processing of the sound of a musician's primary instrument may facilitate timbre-selective absolute pitch. On the other hand, learned differences in timbre and intonation tendencies among the notes afforded by an instrument may provide clues to pitch identification for the highly experienced instrumentalist.

We have suggested that these potentially useful idiosyncrasies in intonation and timbre can be related to three sources of variation in an instrument's sound. First, due to continuous physical differences in the elements of tone production (such as length and thickness of strings and air columns), timbre may vary continuously as pitch increases on a given instrument. Second, instruments may contain categorical timbre variations among different registers (such as the chalumeau, throat, clarion, and extreme or "altissimo" registers on the clarinet; see [24]). Third and finally, timbral and intonational idiosyncrasies for specific pitches may be unique to instrument types (e.g. oboe vs. clarinet, [25]) due to the physical construction of keys, valves etc.; for example, E5 commonly tends to be sharp in intonation on the oboe, but this is not the

case across all instruments. Fitzgerald and Ramsey [25], moreover, demonstrated that B♭4 and C5 belonging to the same register on the oboe show notably different overtone spectra with maxima for the third and sixth harmonics, respectively. Finally, while Snow [26] has cataloged common intonational idiosyncrasies among a number of instruments, the full extent of such differences has not been thoroughly studied. However, musicians become intimately familiar with the specificities of their own instrument type during training and practice.

In our previous paper [12], we proposed the three types of timbre and intonation tendencies that may relate to ISAP, described above, and suggested that the prominence of ISAP among players of a certain type of instrument may be related to the extent to which that instrument displays such tendencies. Given the lack of intonational idiosyncrasies on a modern, equally-tempered piano and the more homogenous modes of tone production across the registral range of pianos and, to a certain extent, string instruments like the violin, there is some reason to believe that both idiosyncratic timbre and intonation may be relatively more prominent on a double-reed, woodwind instrument like the oboe. Yet, specialized practitioners of such instruments have not received the same attention as keyboard and string players in previous research on absolute pitch. We expect that such idiosyncrasies will translate to different degrees among different instruments. Thus, our choice to begin the investigation of ISAP with oboe players is based on the observation that the oboe has a relatively high number of intonational and timbral idiosyncrasies as well as relatively fixed mappings between motor patterns and absolute pitches (unlike piano where the same notes can be played with many different fingerings depending on the context in which they appear).

Our second proposed mechanism of ISAP entails that articulatory motor imagery, as stimulated by the experience of hearing one's primary instrument, facilitates absolute pitch identification. Pitch identification accuracy with one's primary instrument timbre may be due in part to learned connections between sounds and the kinesthetic actions required to produce those sounds [27]. Related bimodal correspondences between absolute pitch categories and visual notation have been demonstrated in AP possessors [28], and internalized motor representations have been related to the widespread ability to identify and produce music at absolute tempo without external reference [29]. In musical practice and performance, musicians constantly connect their motor actions with the sound that is produced [30, 31]. When a musician hears a note played on their primary instrument, motor areas of their brain that are involved in producing that sound are activated; for example, Furukawa, Uehara, and Furuya [32] found that expert pianists demonstrated muscle-specific M1 excitability in response to listening to synthesized piano tones while non-musicians did not (see also [33, 34]). We propose that this kinesthetic memory aids the musician in pitch identification. In the case of wind players, for example, this kinesthetic memory is likely related not only to hand and finger position, but also to embouchure and the tongue, lips, and jaw. Indeed, Choi et al. [35] observed differences in cortical thickness in areas related to the lips and tongue in wind instrumentalists as compared to non-musician controls.

In the two case studies reported by Reymore and Hansen [12], we tested for increased accuracy in pitch identification for oboe tones over piano tones in professional oboists. Of the two oboists tested, one demonstrated ISAP; this oboist's performance provided evidence consistent with the hypotheses that ISAP ability relied on articulatory motor planning and timbral cues. These timbral cues appeared to be driven by pitch-specific idiosyncrasies of the oboe as an instrument type rather than by familiarity with a specific instrument exemplar (i.e., the player's personally-owned instrument) or by familiarity with one's own tone quality. Although the lack of initial supporting evidence for an advantage for one's personal instrument exemplar or style of playing does not disprove that such effects may indeed exist, they are not tested here (see [12] for further commentary).

In this Registered Report Research Article, we report the results of two experiments with a larger sample of professional oboists; both experiments were outlined in the previously published Registered Report Protocol [36]. Experiment 1 tested for the presence and prevalence of ISAP. Having identified individual oboists with ISAP, Experiment 2 tested two proposed mechanisms for ISAP, instrument type-specific timbral/intonational cues and articulatory motor planning, in the sub-group of oboists who demonstrated ISAP in Experiment 1.

For maximal clarity, large portions of the Introduction, Hypothesis, Stimuli, and Procedure sections of the current paper are reproduced with minor modifications from the published Registered Report Protocol [36].

## Experiment 1

### Materials and methods

All experiments involving human participants were conducted with approval from the Institutional Review Board of Danish Neuroscience Centre at Aarhus University/Aarhus University Hospital (DNC-IRB-2020-001). Informed consent was obtained by checking a box on the online platform and confirmed verbally. All statistical analyses were carried out in R (version 3.6.2). Recruitment and data collection took place from 27/03/2021 to 26/06/2021.

**Hypothesis.** Because the goal of Experiment 1 was to identify individuals who exhibited ISAP, the main hypothesis was tested separately for each participant. Specifically, we hypothesized that each individual oboist would demonstrate instrument-specific absolute pitch (ISAP), which we defined as (1) being able to identify pitches played on the oboe with an accuracy that was significantly above chance level assuming octave equivalence (i.e., $1/12 = 8.3\%$), and (2) doing so with an accuracy that was significantly higher for oboe tones than for piano tones. Following our Registered Reports Protocol [36], these hypotheses were tested using a one-sample proportions tests with continuity correction and Pearson's chi-squared tests with Yates' continuity correction.

**Participants.** Participants were recruited online through Facebook groups for oboists/double-reed players and the International Double Reed Society. Email invitations to participate were also sent individually to professional oboists known to one of the authors, who is an oboist.

Sample size for Experiments 1 and 2 was determined through simulations drawing random samples from our previous case-report data (see [36] for further details). A priori, we determined that we would need a minimum of 8 oboists with ISAP in order to proceed with Experiment 2; thus, we established an initial recruitment goal of finding 10 oboists with ISAP through Experiment 1. The plan for recruitment for Experiment 1, outlined in our Registered Report Protocol, was to first collect and assess data from 30 participants. If we did not identify 10 oboists with ISAP within that group, our next step would be to continue recruiting in groups of five, either until we identified 10 oboists with ISAP or reached 50 total participants. This procedure was motivated by the anticipated difficulty of recruiting from a narrow population (expert oboists). However, our recruitment methods were more successful than first anticipated. Thus, in practice, we ran a total of 40 participants in Experiment 1 before assessing the prevalence of ISAP in the sample. Aside from the initial number of participants run, our process followed the process described in the Registered Report Protocol. Specifically, our initial assessment of ISAP among the 40 oboists showed 15 oboists with ISAP (see results below for details); because this number exceeded our a priori goal of 10, we then ended recruitment for Experiment 1.

Participants ($n = 40$, female = 27, male = 12, non-binary = 1) were on average 35.5 years of age (range = 18–65, SD = 13.9). Based on self-report using the single-question assessment from *Ollen Musical Sophistication Index* [37], 28 participants identified as professional

musicians, 8 as semi-professionals, and 4 as serious amateurs. Data on starting age, years of playing the oboe and the piano overall, hours played during the past week, and estimated hours played on average over the past six months are reported in Table 1. These descriptive results confirm a very high level of both long-term training and short-term practice as well as a clear quantitative distinction between oboe as a primary instrument and piano as a secondary instrument.

As AP has previously been linked to training with the fixed-do solfege instruction system [6], in which labels are associated with particular notes, we collected data on participant training and current use of solfege systems. The fixed-do system differs from other common systems, including moveable-do and scale degrees, in which labels are instead assigned based on a pitch's position relative to the tonal center ("tonic") of a piece. Participants reported starting solfege training at an average of 14.1 years of age (range = 6–19 SD = 3.8, not including one participant with no solfege experience and one person reporting starting age of 0). Thirty participants reported training in moveable-do, 25 in scale degrees, and 20 in fixed-do solfege. Many participants reported training in multiple systems—13 participants selected all three options; 10 participants selected two of the options. With regard to current use, 12 participants reported not actively using any solfege system; 17 reported using moveable do, 14 using scale degrees, and 11 using fixed do. Fourteen participants reported currently using two of the three options.

**Stimuli.** As described in detail in the Registered Report Protocol [36], a pilot experiment was conducted on the oboist who demonstrated ISAP in our previous case-report study [12], to test empirically if ISAP accuracy would be even lower for an alternative contrast timbre such as flute or violin, with which she was less familiar, than with piano. Scoring 80.0% correct for oboe tones several months after the original experiment (where she scored 64.4% correct), this oboist demonstrated excellent test-retest reliability. Moreover, despite this participant's experience with piano as a secondary instrument, identification for piano was not more accurate than for either flute or violin, instruments with which she had no experience playing. As a result, we continued to use piano stimuli as a contrast timbre in Experiment 1 (see [36] for further details).

Oboe and piano stimuli were taken from the *McGill University Master Samples* (MUMS) [38]. Both types of stimuli spanned the full range of the 32 available chromatic pitches from the standard range of the oboe from B♭3 to F6. Oboe tones ranged from 2 to 3 seconds in duration and were played at a medium dynamic level with moderate amounts of vibrato. Piano tones were sourced from the "loud" subsample of piano tones with approximately 3 seconds duration played on a 9' Steinway Model D Concert Grand Piano produced at the factory in Hamburg, Germany.

Stimulus preparation and editing for the current experiments were carried out in Cubase 7.0.5. Recorded tracks were segmented into single-tone clips using the Split Function starting

**Table 1. Oboe and piano experience for the 40 participants in Exp. 1.**

|  | Oboe | | | | Piano | | | |
|---|---|---|---|---|---|---|---|---|
|  | *M* | *SD* | *Min* | *Max* | *M* | *SD* | *Min* | *Max* |
| **Years played*** | 23.6 | 13.9 | 6 | 54 | 14.6 | 15.3 | 0 | 58 |
| **Starting age** | 11.4 | 2.1 | 8 | 17 | 9.7 | 5.4 | 4 | 19 |
| **Hours played in past week** | 9.9 | 7.6 | 0 | 30 | 1.5 | 3.2 | 0 | 14 |
| **Average weekly hours played in past 6 months*** | 10.3 | 6.8 | 0 | 25 | 1.8 | 3.0 | 0 | 14 |

*Notes.* *Two participants reported no experience with piano. **Answers for two participants were above 60. As this is not a realistic number for a weekly average, this suggests they misinterpreted the question as asking for a total; these reports were consequently not included in these descriptive statistics.

at 1 second before tone onset with a total clip duration of 5 seconds. All clips were peak-normalized to +18 dB FS (with no pre- or post-crossfading) and exported as MPEG 1 Layer 3 ("mp3") files in stereo with 160 kbps bit rate and 44.1 kHz sample rate. The pitch of each single-tone clip was confirmed using the Android app "Vocal Pitch Monitor" created by Tadao Yamaoka.

**Procedure.** To avoid carryover effects—where superior performance for one instrument could be used to guess the correct pitch of tones played on the other instrument via relative pitch kept in working memory—oboe and piano tones were presented in separate blocks in a reverse counterbalanced order. The use of either oboe-piano-piano-oboe or piano-oboe-oboe-piano was further counterbalanced across participants. All blocks were presented using the Gorilla Experiment Builder (www.gorilla.sc) [39].

Participants met with the experimenter via Zoom video conferencing software. After verifying that participants had headphones and were in a quiet place without distractions or interference from background noise, the experimenter read the instructions out loud while the participant followed along on the screen. Instructions included a description of the system of Scientific Pitch Notation with an accompanying diagram (Fig 1), which was visible to participants throughout the experiment. The participant was given the chance to ask questions and was asked to confirm that they understood. Next, we collected information related to demographics and musical background.

The 32 tones for each instrument were randomly distributed between the two blocks for each instrument, and the 16 tones within each of the four blocks were presented in random order (see Fig 2). Oboists chose each pitch name from a set of all 32 possible pitches with enharmonic equivalents listed when appropriate (e.g., F#4/G♭4; see Fig 3 for screen layout). At the beginning of each block, a picture of either a grand piano or an oboe was displayed to the participant to provide a visual cue (accompanied by text) indicating which instrument would be used in the subsequent block.

After the experiment, participants were asked about response strategies (open-ended), confidence in their own responses (1–7), whether they believed they had ISAP (definitely not / probably not / possibly / probably / definitely), and whether they were aware of the ISAP phenomenon before taking part in the study (yes/no).

## Results

**Pre-registered analyses.** Table 2 and Fig 4 describe the results from the main analysis. For the purposes of illustrating the range of pitch-identification skills, we have categorized participants into two overall categories (ISAP and non-ISAP). Of the 40 participants, 15 (37.5%) met both criteria for ISAP (i.e., above-chance accuracy for the oboe and significantly higher accuracy for oboe than piano tones). Results for individual oboists are included in the S1 Table.

Using subcategories, we made additional distinctions as to whether participants scored statistically significantly above chance for one or both instruments, where the term "quasi"

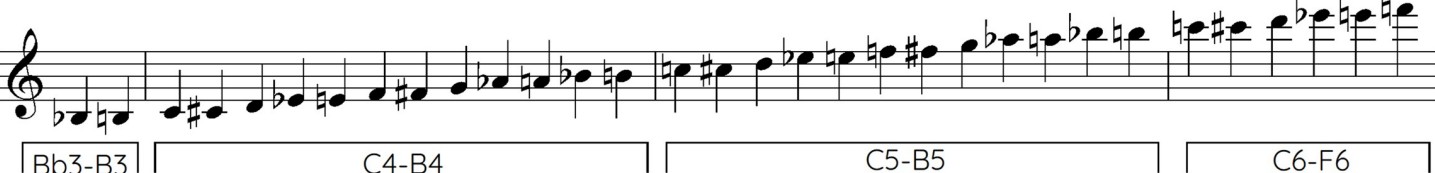

**Fig 1. Range of stimuli.** Diagram depicting the range of pitches from B♭3 to F6 included in the study. This corresponds to the standard range of the oboe (while the oboe can play higher than F6, occurrences of these higher pitches are rare in most musics).

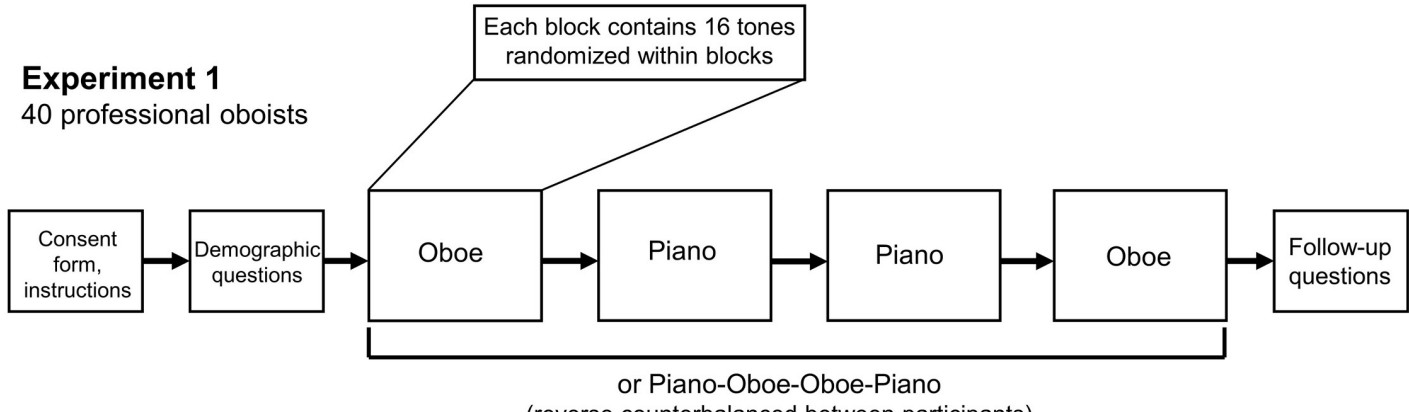

**Experiment 1**
40 professional oboists

Each block contains 16 tones randomized within blocks

Consent form, instructions → Demographic questions → Oboe → Piano → Piano → Oboe → Follow-up questions

or Piano-Oboe-Oboe-Piano
(reverse counterbalanced between participants)

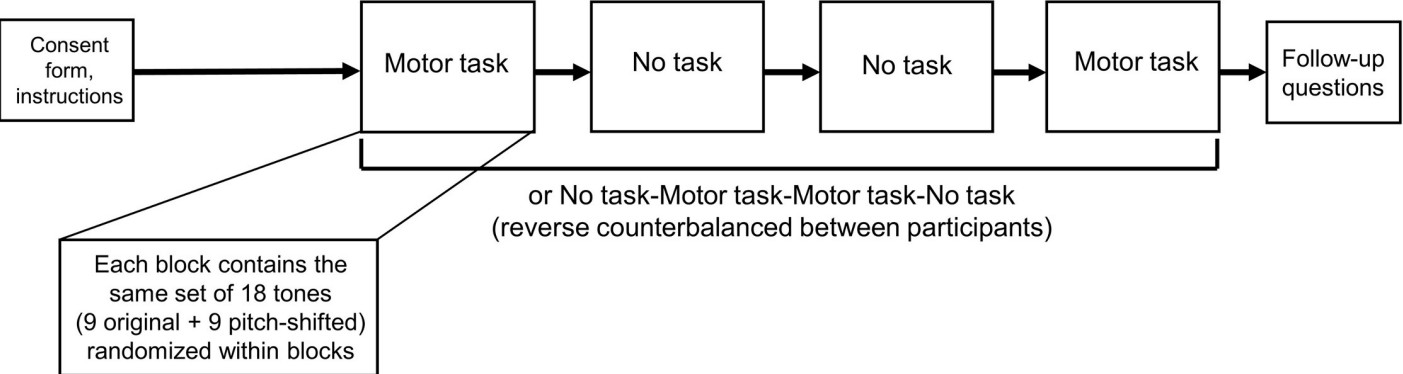

**Experiment 2**
12 participants from Exp. 1 who qualified as having ISAP

Consent form, instructions → Motor task → No task → No task → Motor task → Follow-up questions

or No task-Motor task-Motor task-No task
(reverse counterbalanced between participants)

Each block contains the same set of 18 tones (9 original + 9 pitch-shifted) randomized within blocks

**Fig 2. Experimental design.** Experimental designs for both Experiment 1 and Experiment 2 are illustrated. Both experiments consisted of four principal blocks presented in counterbalanced order, where stimuli within each block were presented randomly.

indicates above-chance performance. Thus, "quasi oboe" describes participants who identified oboe tones above chance but not piano tones. "Quasi both" is used to indicate individuals who scored above chance (i.e., 1/12 = 8.33%) for both oboe and piano tones (but not >90% accuracy overall). An additional distinction of "global AP" is made from the "quasi both" category to distinguish individuals with exceptional levels of accuracy across oboe and piano. While there is no single, agreed-upon threshold for global AP, we followed [13] in considering semitone errors as incorrect responses and categorizing those scoring >90% correct as possessors of global AP.

Overall, 35 participants (87.5%) demonstrated above-chance identification of oboe tones ("quasi oboe", "quasi both", or "global AP") whereas 26 participants (65.0%) did so for piano tones ("quasi piano", "quasi both", or "global AP"). Only one participant demonstrated above-chance performance for piano but not for oboe ("quasi piano").

Further analysis addressed concerns for multiple comparisons and for bias in the estimate of ISAP prevalence in the population of oboists in this experiment. We first considered the probability of none of the participants actually having ISAP, given the observations made in the experiment. Assuming the number of false positives is drawn from a binomial distribution,

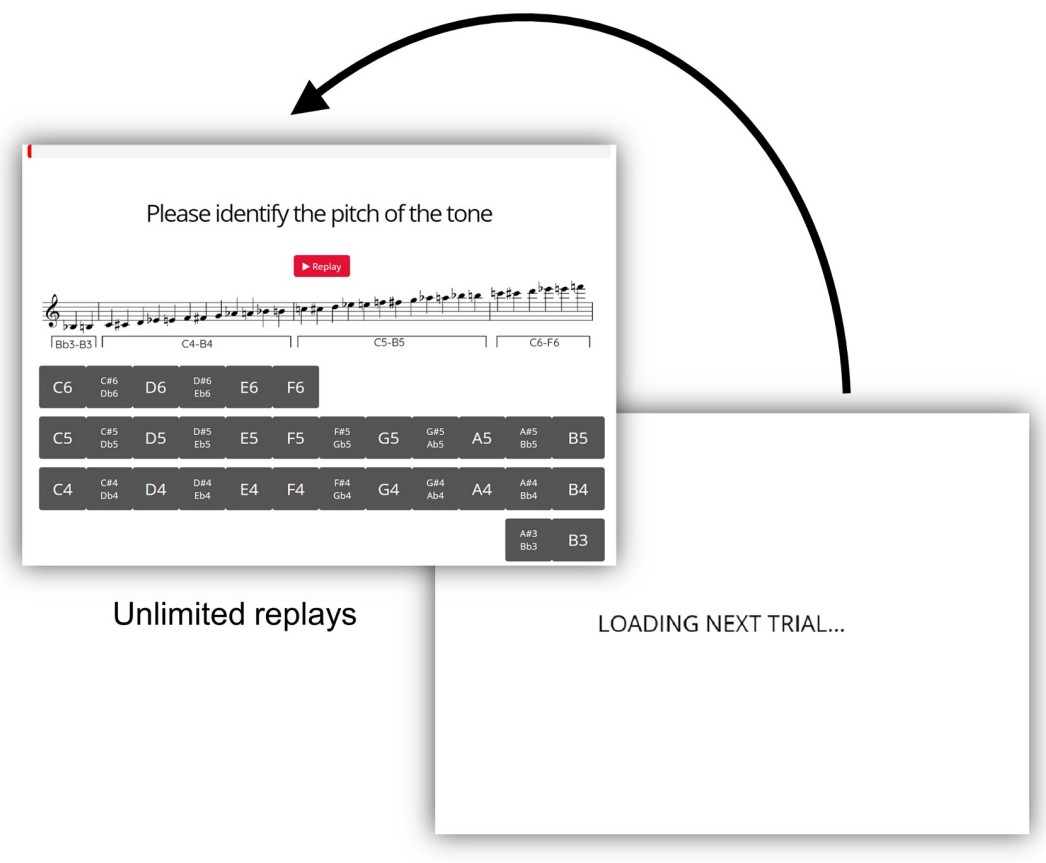

**Fig 3. Trial interface.** The trial interfaces for both Experiments 1 and 2 were identical; participants listened to a tone and selected the name of the note. Participants could replay tones as many times as they wished. Immediately after they responded by clicking a button on the screen using the computer mouse, a loading screen was shown for 500ms to avoid duplicate responses to the same trial.

the probability of no oboists actually having ISAP, given that we observed 15 participants with ISAP out of a sample of 40 oboists, is $P(k >= 15) = .0008$. This result supports rejection of the null hypothesis that there were no participants with true ISAP in the tested population.

Second, to address bias in the observed population prevalence (0.375), we used a Bayesian method [40] to estimate population prevalence from the results of our within-participant tests (*bayesian-prevalence* package in R). Given the observations in Experiment 1, the maximum a posteriori (MAP) estimate—the proportion of the population we would expect to be a true positive for ISAP—is 0.34, (95% HPDI: [0.20 0.50]). This is close to our observed prevalence of 0.375. Here, the highest posterior density interval (HPDI) is a measure of uncertainty, akin to a confidence interval; that is, the probability that the population prevalence is higher than 0.20 is greater than 95%.

**Declared exploratory tests.** *Absolute semitone error.* Difference in mean absolute semitone error between oboe and piano tones was calculated as a participant-level measure of the timbre-specific advantage for oboe tones over piano tones (or vice versa). This measure did not differ significantly between those with and without ISAP ($t(25.69) = -1.464$, $p = .155$) and was also not significantly related to mean response times ($r_s(38) = .146$, $p = .370$) or to the mean number of replays ($r_s(38) = -.021$, $p = .899$) for oboe trials. This difference was also not

**Table 2. Descriptive statistics for Experiment 1.**

| | | n | Gender | Mean age (SD) | Mean accuracy (SD) | | | Mean absolute semitone error (SD) | | |
|---|---|---|---|---|---|---|---|---|---|---|
| | | | | | *Oboe* | *Piano* | *Oboe>Piano* | *Oboe* | *Piano* | *Oboe<Piano* |
| **All** | | 40 | 17F, 12M, 1NB | 35.5 (13.9) | .5859 (.2882) | .4070 (.3258) | 0.1789 (0.1634) | 0.4883 (0.4791) | 1.030 (0.9356) | 0.5422 (0.8462) |
| *ISAP* | *SUB-CATEGORY* | | | | | | | | | |
| **Yes** | **All** | 15 | 11F, 3M, 1NB | 41.4 (17.1) | .6375 (.2014) | 0.2813 (.2299) | .3563 (.0850) | 0.3708 (0.3234) | 1.1729 (0.9802) | 0.8021 (0.9220) |
| | *Quasi ob[1]* | 6 | 5F, 1M, 0NB | 44.2 (18.9) | .4740 (.1159) | .0990 (.0776) | .3750 (.0791) | 0.4635 (0.2633) | 1.4271 (0.7431) | 0.9635 (0.7924) |
| | *Quasi both[1]* | 9 | 6F, 2M, 1NB | 39.6 (16.7) | .7465 (.1708) | .4028 (.2173) | .3438 (.0911) | 0.3090 (0.3592) | 1.0035 (1.1205) | 0.6944 (1.0309) |
| **No** | **All** | 25 | 16F, 9M, 0NB | 32.0 (10.4) | .5550 (.3296) | .4825 (.3475) | .0725 (.0878) | 0.55875 (0.5460) | 0.945 (0.9175) | 0.3863 (0.7748) |
| | *No quasi* | 4 | 2F, 2M, 0NB | 33.3 (13.0) | .1719 (.0313) | .0859 (.0299) | .0859 (.0299) | 1.4219 (0.4554) | 1.2500 (0.6917) | -0.1719 (0.823) |
| | *Quasi ob[1]* | 4 | 3F, 1M, 0NB | 31.0 (14.3) | .2656 (.0541) | .1406 (.0180) | .1250 (.0571) | 0.4453 (0.3098) | 1.4297 (0.6226) | 0.9844 (0.6422) |
| | *Quasi pno[1]* | 1 | 1F, 0M, 0NB | 34.0 (NA) | .1875 (NA) | .2188 (NA) | -.0313 (NA) | 1.6875 (NA) | 4.0312 (NA) | 2.3438 (NA) |
| | *Quasi both[1]* | 11 | 6F, 5M, 0NB | 31.5 (9.9) | .6449 (.2426) | .5511 (.2414) | .0938 (.1027) | 0.3920 (0.2149) | 0.6989 (0.5665) | 0.3068 (0.6012) |
| | *Global AP[2]* | 5 | 4F, 1M, 0NB | 32.4 (10.7) | .9688 (.1159) | .9750 (.0261) | -.0063 (.0464) | 0.1000 (0.1731) | 0.2375 (0.3114) | 0.1375 (0.3981) |

*Notes*: [1]Quasi-absolute pitch for oboe (ob) and/or piano (pno) is defined as scoring significantly (alpha = .05) above chance (i.e., 1/12 = 8.33%), albeit not above 90% accuracy overall. [2]Global absolute pitch is defined as overall accuracy (irrespective of timbre) above 90% (cf. [13])

significantly related to years of experience playing the oboe ($r_s(38) = .306$, $p = .055$) or to recent practice habits ($r_s(38) = -.145$, $p = .374$).

*Response times and number of replays*. A Wilcoxon signed-rank test showed that participants took longer on average when responding to incorrect trials (median of means = 17.9 secs, IQR of means = 14.8 secs) as compared to correct trials (median of means = 13.3 secs, IQR of means = 9.3 secs), $V = 71$, $p < .001$, $n = 40$. Mean number of replays was also higher for incorrect (median of means = 1.68, IQR of means = 0.74) compared to correct trials (median of means = 1.44, IQR of means = 0.58), $V = 96$, $p = < .001$, $n = 40$.

*Co-occurrence of ISAP and AP*. Although it is theoretically possible for global AP and ISAP to co-occur, ceiling effects may obscure such differences. No participants in our sample with global AP (defined as overall accuracy >90%) showed a significant difference between oboe and piano tones. Among individuals with ISAP, 6 (40%) did not score above chance for piano (i.e., "quasi oboe"). Their range for oboe accuracy was 31.25%–65.63% ($M = 47.40\%$, $SD = 11.59\%$). The other nine individuals with ISAP, who did score above chance for piano (i.e., "quasi both") demonstrated a range of oboe accuracy of 53.13%–96.88% ($M = 74.65\%$, $SD = 17.08\%$). That is, while AP and ISAP did not co-occur in our sample, individuals with ISAP who also scored above chance on piano had a significantly higher accuracy for oboe tones ($t(12.97) = -3.682$, $p = .003$).

*Pitch confusions*. Fig 5 illustrates pitch confusions by interval and direction. Inspecting this graph visually, it appears that most errors are within just a few semitones of the correct pitch, with generally fewer confusions made for larger intervals. In the Registered Report Protocol [36], an exploration of whether there might be greater confusion among harmonically related pitches was proposed; that is, whether incorrect responses were more likely to represent

# Pitch−identification accuracy (Exp. 1)

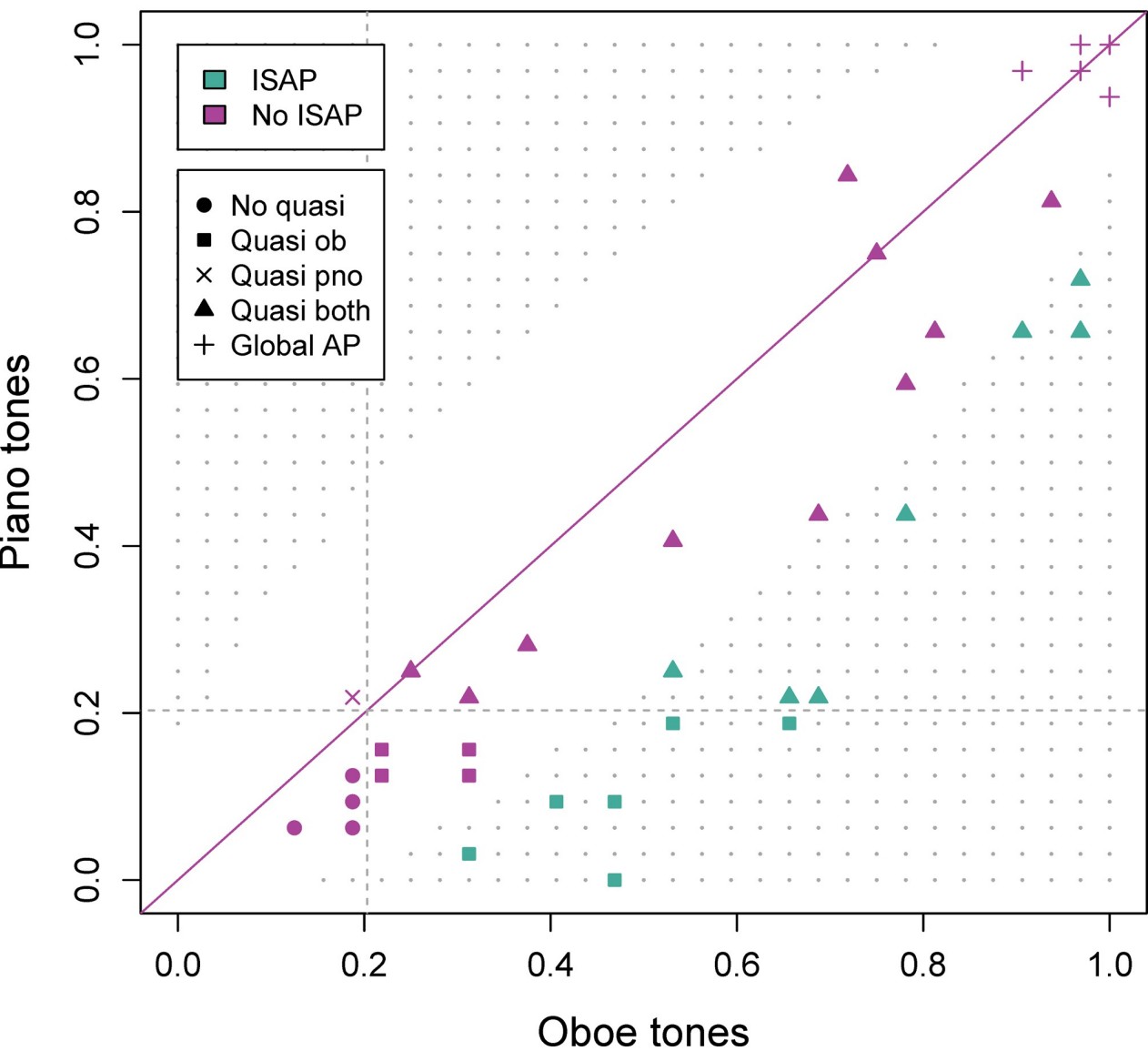

**Fig 4. Mean accuracy (proportion correct) for oboe (ob) and piano (pno) tones for each of the 40 participants in Experiment 1 with color indicating ISAP status and symbol shape indicating membership of the subtypes included in Table 2.** Grey dots represent all hypothetical values for which oboe performance would qualify as significantly better than piano performance (ISAP; bottom right) or vice versa (top left). The dotted lines show thresholds for above-chance performance for piano (horizontal) and oboe (vertical).

harmonically related pitches, such as those an octave, fifth, or major third away from the correct pitch. In addition to proximate confusions, peaks at octaves in either direction and pure fourths and fifths in the descending direction do seemingly tend to occur (Fig 5). An operational theory will need to be formulated to test this possibility systematically in future research.

**Undeclared exploratory tests.** *Accuracy by pitch.* Accuracy for specific pitches varied widely across the range of the oboe (Fig 6). In descriptive terms, people with ISAP exhibited numerically greater variance of accuracy between pitches ($SD = 13.35$) in comparison to

# Pitch confusions

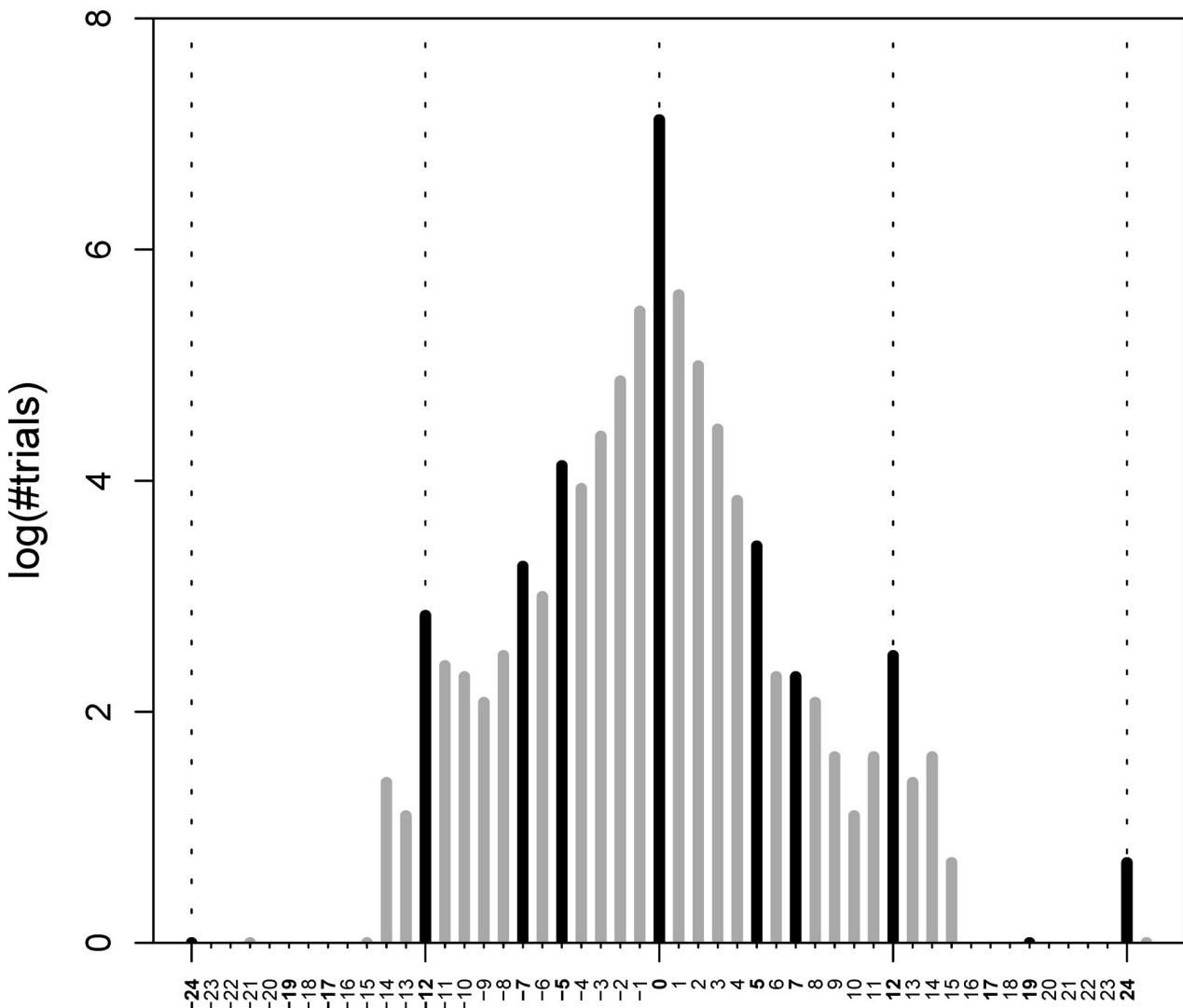

**Fig 5. Number of pitch confusions in all trials from Exp. 1 as a function of semitone error.** Pitch intervals comprising frequencies related by small-integer ratios (i.e., octaves, fifths, and fourths), which were predicted to yield more frequent pitch confusions, are marked in bold font and black color. Given the prominence of correct responses and small-interval pitch confusions, the y-axis is log-transformed for visualization purposes.

people without ISAP (*SD* = 8.29). Specifically, timbrally distinctive notes, such as C5, D♭4, and D♭5 seemed to be identified more accurately by oboists with ISAP. This suggests that the oboists with ISAP had learned to capitalize on timbral idiosyncrasies of specific pitches on the instrument.

Previous work has shown a pitch identification advantage for synthesized piano tones associated with the often-used white keys on the piano keyboard over those associated with the slightly less frequently used black keys [41, 42]. To test for this effect as well as for its potential interaction with ISAP status, a logistic mixed-effects model was fitted predicting correct

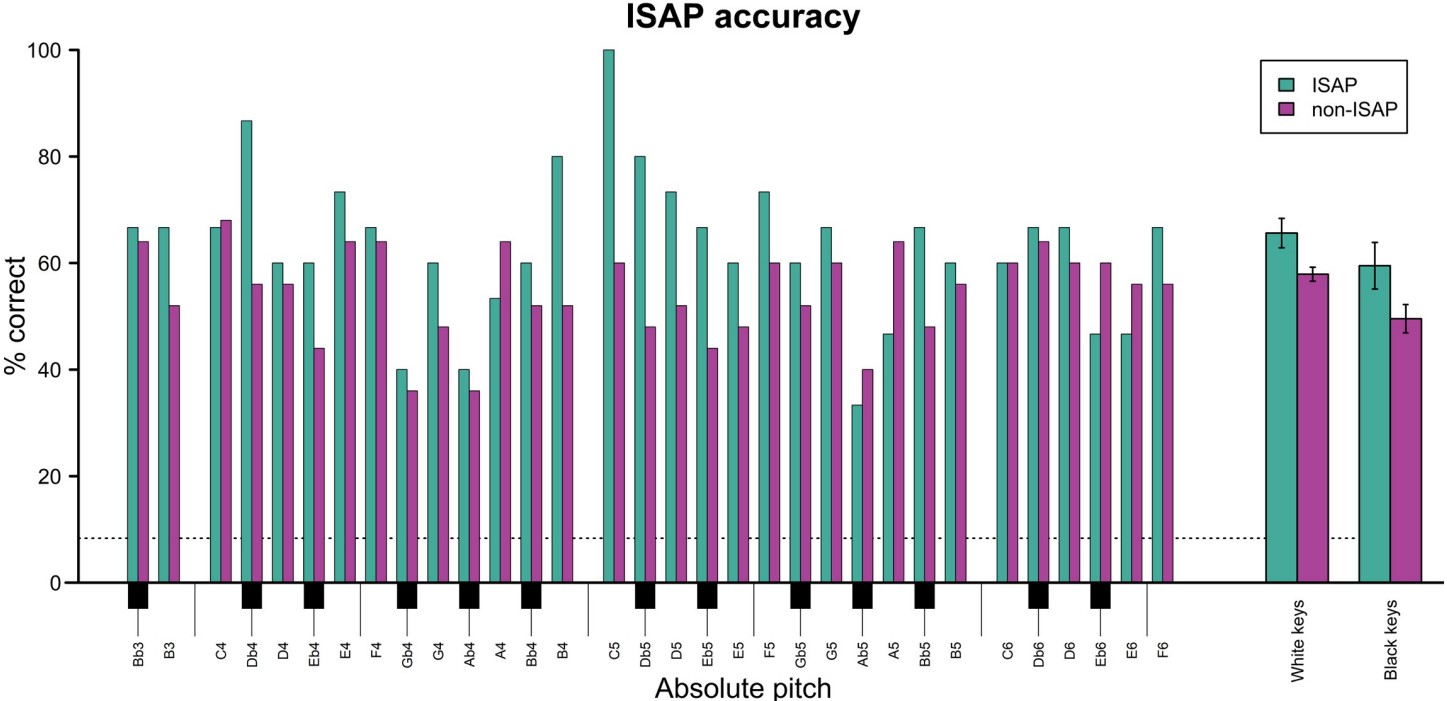

**Fig 6. Pitch identification accuracy for specific pitches for oboe tones played during Experiment 1.** Performance is separated for participants demonstrating significant levels of instrument-specific absolute pitch (ISAP) and those not demonstrating this ability. For the purposes of this plot, participants with global AP were excluded from the latter of these groups as their near-perfect accuracy levels inflate the averages for non-ISAP participants. As evident, in most cases, performance was better overall (and more variable) for people with ISAP than people without ISAP, and performance was generally better for pitches coinciding with white than black piano keys, irrespective of ISAP status.

identification of pitch classes from key color, instrument timbre, ISAP status, as well as interactions for key color with instrument timbre and ISAP status, with random intercepts for participants. Wald tests found significant effects of key color ($Z = -2.774$, $p = .006$) and instrument timbre ($Z = -8.213$, $p < .001$), whereas the interactions for key color-by-instrument timbre ($Z = -.960$, $p = .337$) and key color-by-ISAP status ($Z = 0.880$, $p = .379$) both remained non-significant. That is, in addition to the expected better overall performance for oboe compared to piano timbres, oboists performed significantly better for white-key compared to black-key notes. This effect was, however, not any different between people with and without ISAP or between piano and oboe tones.

*ISAP, demographics, and musical training.* We also investigated potential differences between participants with and without ISAP with respect to demographics and musical training (Table 3). No significant differences between the two groups were observed.

*Post-experiment questions.* When asked whether participants thought they had ISAP, 8 (20%) participants responded "Probably not," while 26 (65%) responded either "Possibly" or "Probably." Only 1 (2.5%) participant selected "Definitely not," while 5 (12.5%) selected "Definitely." Surprisingly, participants with ISAP were not any more likely than participants without ISAP to self-declare as having this ability (see "Self-declared ISAP," Table 3). Of the 40 participants, 36 (90%; 13 people with and 23 people without ISAP) reported that they were aware of the ISAP phenomenon before taking the study).

The principles of grounded theory guided our analysis of response strategy descriptions [43]. Eight principal strategies emerged from data-driven open and axial coding, with several remarks classified into a ninth, miscellaneous category (Table 4). The most common strategy

**Table 3. Exploratory analysis of demographic and music training differences between participants with and without ISAP.**

| | ISAP (*n* = 15) | Non-ISAP (*n* = 25) | ISAP ≠ Non-ISAP | |
| --- | --- | --- | --- | --- |
| | *M (SD)* | *M (SD)* | *Statistic* | *p* |
| Age | 41.4 (17.1) | 32.0 (10.4) | $W = 128$ | .099 |
| Gender | 11F, 3M, 1NB | 16F, 9M, 0NB | $\chi^2(1) = 0.34$ | .559 |
| Self-declared ISAP (1–5) | 3.47 (1.00) | 3.28 (1.06) | $W = 160.5$ | .440 |
| Oboe: | | | | |
| • *starting age* | 11.2 (1.5) | 11.6 (2.4) | $t(37.83) = 0.58$ | .565 |
| • *years of playing* | 29.3 (16.3) | 20.2 (11.2) | $W = 123.5$ | .075 |
| • *hrs last week* | 8.2 (6.8) | 10.8 (8.1) | $t(33.77) = 1.11$ | .275 |
| • *hrs average week* | 10.1 (6.9) | 16.1 (21.4) | $W = 208.5$ | .566 |
| General musical training: | | | | |
| • *starting age* | 7.3 (6.7) | 7.8 (8.1) | $W = 220$ | .370 |

was the use of relative pitch (*n* = 28); however, this strategy was always mentioned in conjunction with one or more other strategies. This will be considered in more detail in the Discussion section below.

Using timbre to identify notes was the second-most common strategy (*n* = 16). Eleven of these participants mentioned timbre or color specifically in conjunction with the oboe. For example, Participant 14 wrote: "For the oboe pitches, I tried to listen to the color of each note. B♭4 and F4 have a covered sound, while B4 and C5 have an open sound." Participant 32 described C5 as "particularly buzzy" and B♭3 as "rambunctious." The theme of timbre was not mentioned specifically in conjunction with the piano except in comparison with the oboe: for example, Participant 35 wrote: "For the oboe notes, the color of some of the pitches were very easy to recognize; this was much harder for the piano pitches."

Motor imagery (*n* = 6) was only referenced in relation to the oboe. For example, Participant 4 wrote: "For the oboe ones, it helped (?) [sic] to think about the fingerings for different notes and see if it felt right." Participant 13 described their strategy as "[f]eeling where I thought the note would be on the oboe. (There's a physical feeling connected to most notes at this point.)"

Other strategies described included the use of instinct or a sense of familiarity (*n* = 8), identification of register (*n* = 10), using a memorized note as an anchor (*n* = 7), recalling pieces with known pitches in order to establish an anchor pitch (*n* = 9), and relating the heard pitches to their vocal range (*n* = 6). A logistic regression model showed no systematic relationship between strategy and whether participants exhibited ISAP (all $Zs \leq 1.316$, $p \geq .188$).

**Table 4. Strategies emerging from qualitative analysis of post-experiment questions.**

| Strategy | Number of Participants | | |
| --- | --- | --- | --- |
| | *ISAP* | *No ISAP* | *Total* |
| Relative pitch | 12 (80%) | 16 (64%) | 28 (70%) |
| Timbre | 7 (47%) | 9 (36%) | 16 (40%) |
| Register | 2 (13%) | 8 (32%) | 10 (25%) |
| Reference to piece | 4 (27%) | 5 (20%) | 9 (23%) |
| Instinct/Familiarity | 4 (27%) | 4 (16%) | 8 (20%) |
| Using a memorized note as an anchor | 0 (0%) | 7 (28%) | 7 (18%) |
| Motor imagery | 3 (20%) | 3 (12%) | 6 (15%) |
| Relating heard pitch to vocal range | 3 (20%) | 3 (12%) | 6 (15%) |
| Miscellaneous | 3 (20%) | 1 (4%) | 4 (10%) |

## Experiment 2

Experiment 2 was designed to investigate the potential underlying mechanisms of instrument-specific absolute pitch (ISAP). Specifically, we tested whether our case-report findings that ISAP identification is subject to deterioration from artificial pitch-shifting and a motor interference task [12] would generalize to a wider population of expert oboists with established ISAP ability. Recruitment and data collection took place from 18/07/2021 to 25/09/2021, and ethics approval and consent procedures were the same as for Experiment 1.

## Materials and methods

**Hypotheses.**

**H1.** Oboists with ISAP will be less successful in identifying the pitch of artificially pitch-shifted oboe tones than the pitch of non-transposed oboe tones.

**H2.** Oboists with ISAP will be less successful in identifying the pitch of oboe tones in the motor interference condition as compared to the condition with no motor interference.

For the purpose of testing these hypotheses, we operationalized pitch identification success in two ways: (a) accuracy (i.e., proportion of correct responses) and (b) precision (i.e., absolute semitone error values).

**Participants.** Participants in Experiment 2 ($n$ = 12; female = 9, male = 2, non-binary = 1) were recruited from the subset of participants in Experiment 1 who demonstrated ISAP ability. As for Experiment 1, the sample size was determined through simulations drawing random samples from our previous case-report data (see [36] for further details). Participants were 44.0 years old on average (range 18–65, SD = 17.6). Responding to the single-measure self-identification question from the Ollen Musical Sophistication Index [37], nine participants identified as professional musicians and three as semi-professional musicians. Participants had been playing oboe for an average of 31.6 years (range = 8–54, SD = 16.9). Within the week prior to the experiment, participants reported having played oboe for an average of 10.8 hours (range 0–24, SD = 9.1). When asked to estimate their average weekly practice over the past six months, participants reported an average of 9.7 hours (range 0–22, SD = 7.7).

**Stimuli.** The oboe tones from the MUMS soundbank [38] used for Experiment 1 were also used in Experiment 2. Sound clip segmentation, peak-normalization, and file formats are thus as described above.

For the purposes of generating the pitch-shifted stimuli, the Pitch Shift function in Cubase 7.0.5 was used to manipulate a copy of each sound clip. Specifically, pitch was shifted up or down according to a pattern by which every consecutive set of eight pitches is transposed by +4, -1, +3, -2, +2, -3, +1, and -4 semitones. This process resulted in a new set with the same sounding pitches as the original one (see Table 1 in the Registered Report Protocol [36]) and ensures that transpositions differ between consecutive octaves. Pitch shifting used the Time Correction setting to ensure that the duration of each clip stayed the same as well as the Solo Musical setting, which uses a high-quality algorithm optimized for offline processing of monophonic musical material. Formant Preservation was not applied because this setting generated clearly audible artefacts manifesting as background "whirling" noises before and after the oboe tones when preparing stimuli for a previous experiment [12].

**Procedure.** To test for effects of motor imagery on ISAP, we developed an interference task that was expected to impair pitch-naming accuracy because it increases demands on motor-related brain areas involved in playing the instrument. In the case of the oboe, this

includes the hands and fingers as well as lips and jaw, which are called upon for crucial embouchure adjustments while playing.

Motor tasks may interfere with auditory imagery; for example, Beaman, Powell, and Rapley [44] found that chewing gum has negative effects on spontaneous musical recollection (earworms), in support of the idea that chewing gum interferes with motor-related subvocalization or subvocalization-like processes that are linked to earworms. Consequently, a motor interference task was implemented in Experiment 2. Although the gum-chewing task was used in a prior case-study experiment [12], in which the experimenters were physically present to provide the gum, this method could not be employed in the current experiment, as participants performed the task virtually and were not expected to have gum readily available. Furthermore, it would be difficult to assess compliance with the gum-chewing task via videoconferencing software. Instead, we agreed that a comparable task would be to hold a pen or pencil between the teeth. While this does not involve movement like gum-chewing does, the position does interfere with the ability to manipulate the lips, tongue, and embouchure-related muscles, and we were able to easily assess via Zoom whether or not the participant complied with the task.

Experiment 2 comprised a full factorial design crossing the following two two-level factors: transposition (original vs. pitch-shifted), and motor interference (no interference vs. motor interference). Transposition, in this regard, refers to the fact that half of the presented tones had been pitch-shifted as described in the Stimuli section above. Whereas the no interference condition entailed no further task in addition to pitch identification, the motor interference condition entailed two concurrent tasks which were performed by the oboists while listening to the stimuli and identifying pitches. Specifically, in this condition, oboists were asked to hold a pen or pencil horizontally between their teeth and to continuously wiggle the fingers of their left hand. To remind participants which motor condition was in play during the experiment, images depicting the motor/no motor condition were included before each block as well as during all trials in the upper right corner of the experiment interface. These images contained clip art of a woman holding a pencil between her teeth and a man wiggling the fingers of his left hand with a plus sign in between and a version of the same images grayed out and crossed out.

Based on the experimental design and simulations using the case-report data [36], Experiment 2 consisted of four blocks of 18 trials each (Fig 2). Overall, each participant judged a total of 36 unique sound files, each presented twice (with and without motor interference) for a total of 72 trials. The 36 unique sound files consisted of 18 different tones presented at their original pitch level as well as the pitch-shifted versions of each of these 18 tones. During two of the four blocks, participants were instructed to engage in the motor interference task. Participants were not aware that half of the tones were artificially pitch-shifted. The 36 unique stimuli were randomly distributed across the two blocks of the motor interference condition and across the two blocks of the no motor interference condition, and the 18 stimuli in each block were presented to each oboist participant in random order. Experiment 2 was also created and hosted using the Gorilla Experiment Builder (www.gorilla.sc) [39].

The 18 pitches tested for each oboist were selected with respect to their individual performance in Experiment 1; specifically, oboists were tested on the 18 pitches for which they demonstrated the highest accuracy (with ties determined by random selection). While ideally we would have tested the full range of the oboe for each participant, given the number of conditions, this would have resulted in an excessively long experiment. By selecting the pitches for each participant that yielded the highest accuracy in Experiment 1, our aim was to increase the likelihood of observing the predicted interference in accuracy. The subset of pitch-shifted tones were matched in original pitch to the base set of tones selected by means of previous success. This approach avoided a scenario in which the same set of 18 pitch names were correct within each of the four blocks. This, if noticed by the participants, could have provided a subtle

cue influencing their responses in later blocks. This risk was further minimized through the random distribution of the 36 unique sound stimuli across the two blocks of each type. Due to a technical failure, one participant was exposed to the pitches intended for another participant. However, as 50% of the peak performance pitches for these two individuals were identical, we decided to include all data from this participant in the analysis.

After each block, two validity check questions confirmed that participants always listened through headphones and complied with task instructions in terms of holding a pen or pencil between their teeth and moving their left fingers in the motor interference conditions or refraining from doing so in the conditions without motor interference. Participants completed the same post-experiment questions as described for Experiment 1.

The four blocks of Experiment 2 were presented in a reverse counterbalanced order for each participant (Fig 2). With two conditions—motor interference (M) and no interference (N)—the two orders (M-N-N-M and N-M-M-N) were used an equal number of times across the participants in Experiment 2.

### Results

**Pre-registered analyses.** Using the glmer() function from the *lme4* package, we built a logistic mixed effects model predicting pitch identification (correct or incorrect) with fixed binary effects for transposition and motor conditions, plus a random intercept term for participant (Fig 7). To test significance of each of these conditions, likelihood ratio tests were performed using the lrtest() function from the "lmtest" package. The effect of transposition was significant ($X^2$ = 4.8411, $p$ = .028); participants were more likely to identify artificially transposed pitches incorrectly as compared to untransposed pitches. The effect of motor interference was not significant ($X^2$ = 0.0248, $p$ = .875).

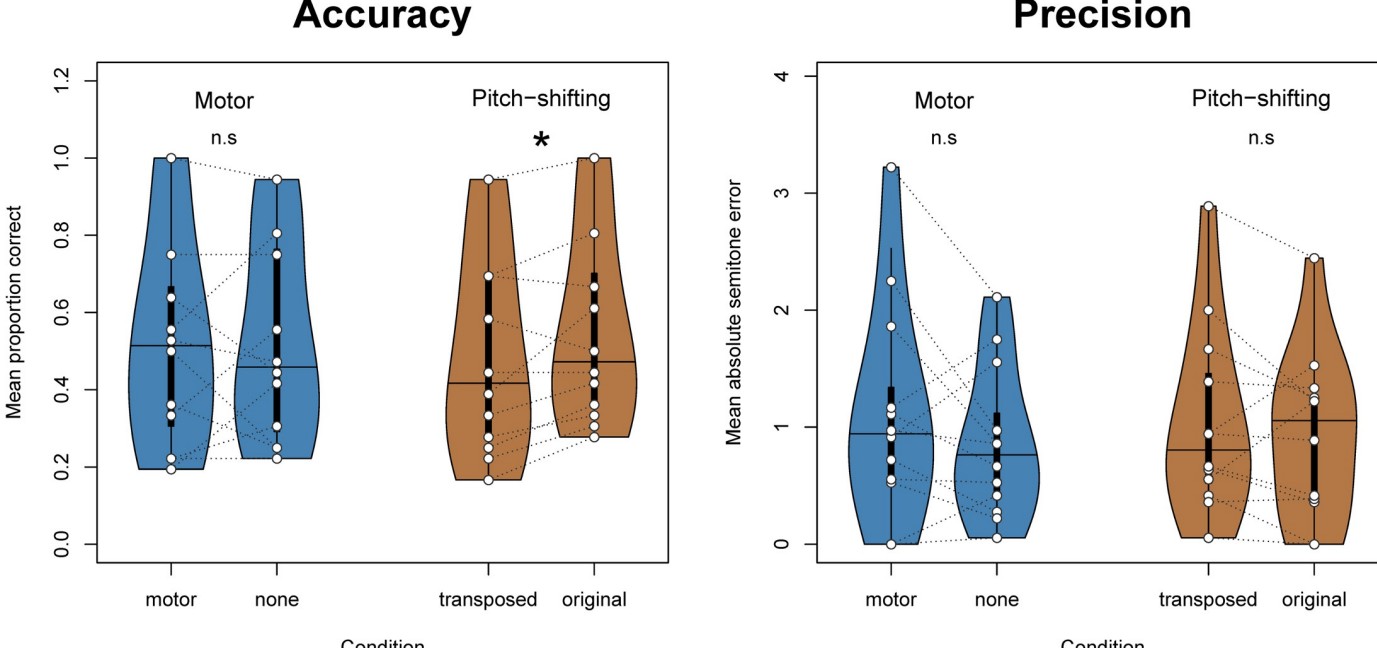

**Fig 7. Violin plots of accuracy and precision in terms of mean percentage of correct responses and mean absolute semitone errors, respectively, for each of the 12 participants in Exp. 2.** Only artificial pitch-shifting (not the motor interference task) impaired performance, with a significant effect for accuracy but not for precision. Vertical lines indicate the median of participant-wise means, and the tops and bottoms of the colored areas indicate the upper and lower adjacent values, respectively. Means for individual participants are depicted with white dots connected with dotted lines in dark gray.

To model the degree of variation around the correct pitch value, a cumulative link model was fitted to absolute semitone error values using the clmm() function from the *ordinal* package. Terms again included transposition and motor interference as fixed binary effects plus a random intercept for participant. In order to obtain model convergence, absolute error values larger than 6 semitones had to be excluded. Likelihood ratio tests, here, showed no effects of transposition ($X^2$ = 1.7011, $p$ = .192) or motor interference ($X^2$ = 1.1871, $p$ = .276).

**Declared exploratory tests.** *Response times and number of replays*. As in Experiment 1, participants spent longer time on average for incorrect trials (mean of means = 18.7s, SD of means = 8.1s) as compared to correct trials (mean of means = 14.7s, SD of means = 7.8s), $t$(11) = -6.970, $p$ < .001. Mean number of replays for stimuli was also higher for incorrect trials (median of means = 1.69, IQR of means = 0.61) than for correct trials (median of means = 1.32, IQR of means = 0.57), $V$ = 9, $p$ = .016, $n$ = 12. However, no significant correlation was observed between overall accuracy and mean response time ($r_s$(10) = -.2386, $p$ = .455) or mean number of replays ($r_s$(10) = -.2377, $p$ = .457) for each participant across correct and incorrect trials.

*Modeling transposition as a continuous variable*. In the case-report data in [12], we found in secondary analysis that modeling pitch-shifting as a continuous variable (i.e., the absolute number of semitones shifted) rather than a binary variable improved the model slightly and lowered both the Akaike and Bayesian Information Criteria. Proper hypothesis-testing would require a research design optimized for answering this question; however, here, we declared an exploratory test along the lines of [12], for the purpose of determining whether future research should ideally operate with pitch-shifting as a continuous variable. This analysis showed no significant effects of absolute pitch-shifting (estimate = .0987 [95% CI: -.0955, .2940], $z$ = 1.00, $p$ = .316) or of motor interference (estimate = .04967 [95% CI: -.3876, .4876], $z$ = 0.224, $p$ = .823). Thus, pitch-identification accuracy did not appear to depend on the extent of pitch-shifting applied.

**Undeclared exploratory tests.** *Direction of pitch-shifting and semitone error*. If our participants are indeed using timbre as a cue for pitch labelling, as theorized in our H1, then we would expect them to show a tendency to guess the original pitch of artificially pitch-shifted tones. Indeed, out of the 219 incorrect, pitch-shifted trials, participants erroneously guessed the original pitch in 45 trials (20.5%). However, because we know that not all incorrect responses are equally likely (cf. Fig 6), we cannot use the equiprobable chance level of 8.33% as a valid null hypothesis here.

Thus, for inferential tests of our prediction, we instead considered that the hypothesized tendency for guessing the original pitch would moreover result in a negative association between the direction of pitch-shifting and the direction of semitone error. That is, on incorrect trials, if a tone was pitch-shifted upwards, we would expect participants to underestimate pitch, and if a tone was pitch-shifted downwards, we would expect them to overestimate pitch. To test this approximation of our main conjecture, we observed that out of the 219 incorrect, pitch-shifted trials, those that were transposed upwards ($n$ = 109), 72.5% ($n$ = 79) were underestimated whereas 27.5% ($n$ = 30) were overestimated. Out of those that were transposed downwards ($n$ = 110), 44.5% ($n$ = 49) were underestimated whereas 55.5% ($n$ = 61) were overestimated. Consistent with our expectation, a chi-squared test with continuity correction found these proportions to differ significantly ($\chi^2$(1) = 16.456, $p$ < .001).

*Accuracy and demographic/musical variables*. Spearman's correlations were used to investigate potential relationships between accuracy and demographic and musical training variables. The number of hours participants reported playing oboe within the previous week was significantly correlated with their overall pitch-identification accuracy ($r_s$ = .7342, $p$ = .007), as was

the number of hours reported playing oboe per week on average over the course of the previous six months ($r_s$ = .6496, $p$ = .022). No other significant correlations were observed.

## Discussion

This study provides empirical tests of a theory of instrument-specific absolute pitch (ISAP), referring to the higher accuracy in absolute pitch identification for one's own musical instrument type over other instruments. ISAP ability has previously been identified in small samples of expert musicians but has not yet been formally established on a larger scale. In Experiment 1, 37.5% of the 40 oboist participants demonstrated ISAP; that is, they both identified oboe tones with above-chance accuracy and performed significantly better for oboe than for piano tones. Generally speaking, we observed a spectrum of pitch identification abilities for both oboe and piano tones separately and for the advantage of one timbre over the other. Results from Experiment 2 supported the hypothesis that oboists with ISAP would be less accurate in identifying the pitch of artificially pitch-shifted oboe tones than of non-transposed oboe tones. However, oboists were not significantly less successful in a motor interference condition (during which they held a pencil between their teeth and wiggled the fingers of their left hand) than in the condition with no motor interference. No effects of either transposition or motor interference were observed on the less sensitive, precision-related measure of absolute semitone error.

Notably, although 15 participants fulfilled both criteria for ISAP, 32 of the 40 participants performed numerically better on oboe tones than on piano tones, while only 4 performed numerically better for piano. Thirty-five participants scored significantly above chance for oboe tones, and 26 scored significantly above chance for piano tones—yet, these populations overlap almost completely, as only one participant scored significantly above chance for piano but not for oboe. That is, although the difference between oboe and piano accuracy was not significant for participants in the non-ISAP group (Table 2), accuracies for the two instruments were highly asymmetrical (i.e., it was rare for an oboist to score better on piano than on oboe). Even among the non-ISAP group, the average superiority in accuracy for oboe over piano tones was seven percentage points. Along with the observation that differences between oboe and piano accuracies were distributed along a continuum (rather than bimodally, see Fig 4), this suggests that effect size for ISAP varies among oboists (as demonstrated by different individual distances from the diagonal in Fig 4). That is, some with ISAP show larger differences in accuracy between oboe and piano than others; the larger the difference, the fewer trials are needed to detect ISAP with our experimental design. Thus, we speculate that with more trials per participant (i.e., by increasing statistical power), more oboists may have qualified as having ISAP in Experiment 1. As we found that the pitch identification portion of both experiments took our participants less time than we had anticipated, more trials could feasibly be included in future experiments.

Although further investigation is needed to map the range of accuracy differences, our data are consistent with the widely supported continuum of pitch-identification abilities [45–48]. We also observed ISAP ability both in oboists with low and high baseline levels of pitch identification accuracy. With this consideration, future experimental designs may benefit from considering ISAP continuously, rather than categorically, and assessing individual differences in ISAP ability.

In case-report data [12], we observed that an oboist with ISAP showed significantly lower absolute semitone error for oboe tones than for piano tones—that is, incorrect answers tended to be closer to the correct pitch for the oboe than for the piano. This pattern was not found for the oboist without ISAP. Thus, we hypothesized that the absolute semitone error difference

between oboe and piano tones would be significantly greater for those with ISAP than for those without ISAP. However, results of Experiment 1 did not support this hypothesis: although the difference was numerically in the predicted direction (mean absolute semitone error advantage for oboe over piano tones was 0.802 for those with ISAP and 0.386 for those without), this difference was not significant. The effects of pitch-shifting on accuracy in Experiment 2 also did not manifest as significant effects on precision. Firstly, it may be the case that the case-report findings are simply not generalizable to a larger population of oboists. Secondly, failure to observe the effect may be related to individual differences in effect sizes, as discussed in the previous paragraph, whereby participants with relatively low degrees of ISAP are mischaracterized as not having ISAP. These potential false negative cases, included in the semitone error analysis as part of the non-ISAP group, may have obscured the observability of a difference in absolute semitone error between the groups. Thirdly, as suggested by simulations based on our previously published case-report data [36], our experimental design may not have entailed sufficient statistical power to detect precision effects. Taken together, these results suggest that precision may be a less sensitive measure of instrument-specific absolute pitch than accuracy. This limited sensitivity for absolute semitone error as a precision measure may in turn apply to studies of global absolute pitch. Yet, if used in future studies, absolute semitone error may still reveal important information—for example, about response strategies such as the reliance on identifying the correct register but not necessarily the correct absolute pitch [12].

A key goal of our study was to test causally whether ISAP ability relies on timbral idiosyncrasies and/or motor imagery. The results of Experiment 2 support the former but not the latter proposed mechanism. Specifically, pitch-identification accuracy for oboists with ISAP was significantly impaired by artificial pitch-shifting but not by the motor interference task. In an exploratory test, we further found that ISAP possessors showed a tendency to overestimate the pitch of tones that were shifted downwards and underestimate the pitch of tones that were shifted upwards. This provides converging evidence for our timbre-related hypothesis.

We have previously proposed three types of timbral variation which may aid instrumentalists in identifying pitch: continuous variation in modes of tone production across the instrument's pitch range, characteristics related to individual registers (categorical), and idiosyncrasies of individual pitches (such as the infamously nasal C5 on the oboe) [36]. Although the current experiments were not designed to disentangle these potential sources of variation, both register and individual pitches were discussed by participants when describing their strategies. For example, one participant noted that "I used cues from the timbre of the sound to choose what octave range it might be." As described in the Results section, participants also referred to pitches with particular characteristics, for example, describing C5 as "open" or "buzzy." Comments regarding the use of register recall a recent study [49], which found that musicians were able to distinguish between equivalent cello pitches played on high- and low-register string locations. The authors also observed differences in spectral acoustic features between registers. Future research might consider whether there is an instrument-specific advantage for the identification of string register—that is, whether cellists (or string players in general) are more adept at identifying string register than non-cellist (or non-string playing) musicians.

Evidence of increased accuracy for timbrally idiosyncratic notes is apparent in Fig 5. In general, the notes demonstrating the greatest differences in accuracy between those with and without ISAP are among the oboe's most timbrally distinct notes. For example, C5 was indeed identified correctly by 100% of ISAP oboists but only by 60% of non-ISAP oboists. This is consistent with its reputation as the oboe's most distinctive note, identifiable because of its relatively high brightness and nasality, as mentioned in multiple participant comments.

Anecdotally, D♭4 is typically more fuzzy or muffled in comparison to other notes within the same register and feels more resistant to play; D♭5 also reflects these qualities but is furthermore distinguishable from D♭4 because it is higher in register and therefore less dark in tone color. These two tones were amongst the ones showing the greatest difference in accuracy between oboists with and without ISAP. Conversely, notes with relatively low identification accuracy may tend to be less timbrally distinctive. For example, G♭4 and A♭4 are relatively consistent in timbre with their neighboring pitches, potentially resulting in greater confusion among notes close in pitch in this register. Interestingly, the lower accuracy for these notes appears to dominate responses from both those with and without ISAP, whereas the higher accuracy for more timbrally distinct notes as discussed above appears only to be the case for those with ISAP.

Although higher accuracy for A4 might be expected based on its familiarity as an orchestral tuning pitch, as suggested by participant comments and results form our prior study [12], we did not observe such a trend among either group. Deutsch, Le, Shen, and Li [50] also did not observe any special status for the orchestral tuning pitch among a larger population of musicians. Yet, seven in the current study mentioned that they felt they were able to identify or produce A4. This adds evidence to the observed disconnect between subjective self-assessment and actual pitch identification performance which was also evident from the present participants' inability to identify whether they had ISAP or not.

Consistent with previous findings (e.g., [16, 50]), participants did, however, identify pitches associated with the black keys of the keyboard (e.g., C#, E♭) less accurately than those associated with the white piano keys (e.g., C, E). The current study demonstrated that this white-key advantage did not interact with either ISAP ability or instrument timbre (piano vs. oboe). This apparent lack of interaction between key color and instrument timbre suggests that statistical learning of pitch could provide a more convincing explanation for this advantage, irrespective of timbre [42, 51, 52]. Yet, the fact that Schlemmer et al. [19] did not find this white-key advantage for unfamiliar instruments (which participants had not taken lessons on themselves) hints at the possibility that piano was perhaps too familiar for our participants to show this timbre-related difference. More tailored experiments are needed to fully resolve this question. As the effect did not vary with ISAP status, however, the mechanism(s) driving the white-key advantage may be used by both those with and without ISAP. Future corpus studies of instrument-specific musical repertoires could provide more robust tests of the role of statistical learning in ISAP ability.

With respect to the lack of support for the motor hypothesis, it is possible that articulatory motor planning is not a mechanism of ISAP, that it is not a common enough mechanism to have been observable in our sample, or that this mechanism is less effective for oboists than for other instrumentalists. However, it is also possible that the effect was not observable for other reasons. First, due to the virtual experimental conditions (via Zoom), the motor task was modified in the current version of the experiment as compared to the case-report study [12]: participants held a pencil between their teeth rather than chewing gum, and they were required to wiggle the fingers of only one of their hands, rather than two. It is possible that the lessened activity was not sufficient to interfere with the purported motor planning mechanism.

Moving on to the exploratory results, we did not observe significant relationships of ISAP with any of the demographic, musical training, or practice-related variables. This is in contrast to previous research on global absolute pitch, which has often linked this ability to early training [6, 11, 53]. This, in turn, suggests either that separate mechanisms underlie global AP and ISAP, or that more sensitive research designs tailored towards testing the specific role of early training in ISAP ability are needed. We also did not observe ISAP among participants with global AP (defined here as having overall accuracy above 90%). Ceiling effects may prevent detecting instrument-specific advantages in people who already possess exceptionally high

general AP ability, making it challenging to identify whether ISAP and AP can co-occur and if they can, whether the underlying neurocognitive mechanisms are distinct or overlapping. Future studies might consider excluding participants with global AP if, for example, the goal is to detect individuals with ISAP. On the other hand, a more in-depth study of whether aspects of ISAP do exist among those with global AP may provide additional insight; however, because of the potential difficulty of detecting a difference in accuracy when overall accuracy is already near or at ceiling, high statistical power or alternative measures may be needed.

After the concept of ISAP was explained to the participants at the end of Experiment 1, the large majority of participants (90%) answered that they had been previously aware of the phenomenon, and 78% of participants felt that it was possible they had ISAP. However, oboists with ISAP were not generally more likely to think that they had ISAP than those without, suggesting that although many oboists were aware of the phenomenon, they were not likely to estimate well whether they personally had ISAP. Of course, cases in which participants without significant ISAP erroneously expressed belief that they did have ISAP may relate to the statistical power of the study. Firstly, participants with high levels of overall pitch identification accuracy would not have demonstrated ISAP due to ceiling effects. Secondly, participants with low levels of ISAP (i.e., smaller differences in accuracy between oboe and piano tones) that were below the detectable difference may also not have shown significant differences in accuracy in the context of this study.

In examining the self-declared pitch identification tactics of participants in Experiment 1, we found that both timbre and kinesthetic strategies were prominent. Consistent with our quantitative results in Experiment 2, timbre was a commonly-described strategy, mentioned by 40% of participants. In most of these cases, timbre was mentioned specifically in conjunction with the oboe rather than the piano, and when piano timbre was mentioned, it was described as less useful than oboe timbre. On the other hand, although we did not find an effect of motor interference, strategies related to motor imagery were described by 15% of participants. Because of the free-response paradigm, which was implemented as an exploratory measure, we are not able to draw definitive conclusions about potential differences in strategies reported by those with and without ISAP. Yet, the fact that both timbral and motor strategies were described by ISAP possessors and non-possessors is consistent with the observed continuity of instrument-specific pitch identification ability. These findings can be used to inform future experiments to more systematically assess strategy.

The prominence of relative pitch as a strategy among both ISAP and non-ISAP oboists also merits discussion. Of course, relative pitch cannot be used successfully without the presence of a known reference pitch. As we previously argued [36], it is possible that individuals may have memorized a particular pitch, such as the tuning 'A4,' and are able to auralize and use this pitch as a reference pitch (as was also proposed by Bachem in 1955 [54])—however, we note that this strategy should likely work equally well with any timbre. In Experiment 1, the strategic use of such an anchor note was mentioned by seven participants. However, none of those participants exhibited ISAP. Participants who specifically described the use of relative pitch also mentioned other strategies which could plausibly be linked to finding a reference tone, including recognition based on instinct/familiarity or on association with a piece with a known starting pitch. Previously [36], we remarked that timbrally-informed identification of a particular note from the primary instrument may result in ISAP via a mixture of instrument-specific absolute and relative pitch identification strategies. In general, participants described multiple strategies, and relative pitch was never listed as a participant's only strategy. This suggests that many oboists may use a mix of absolute and relative pitch strategies, where ISAP provides an advantage as a source of reference tones, for example, via timbrally idiosyncratic pitches on the instrument in question.

## Conclusions

The current study for the first time suggests that instrument-specific absolute pitch (ISAP) occurs in the wider population of highly trained oboists, with an approximate prevalence of one third. We observed that artificially pitch-shifting tones significantly interfered with pitch identification accuracy among oboists with ISAP, suggesting that ISAP may rely in part on timbral idiosyncrasies typical of a specific instrument. This provides a possible path forward for future studies extending the scientific understanding of the ISAP phenomenon to other instrument types, expertise levels, and musical contexts. The ISAP research program represents an important next step in a long-standing development in the investigation of absolute pitch, progressing from abstract sine tones towards more ecologically valid stimuli with greater spectrotemporal complexity (e.g., [55]). This work may help further clarify the role of timbre in pitch identification as well as elucidate the complex relationship between pitch and timbre perception in music and beyond [56]. More generally, understanding ISAP may also contribute to a deeper knowledge of specialized expertise, representing a wide range of implicit abilities that are not addressed directly in training, but which may develop through practice of a related skill set.

## Supporting information

**S1 Table. Participant-level descriptive statistics for Experiment 1.**
(DOCX)

## Author Contributions

**Conceptualization:** Niels Chr. Hansen, Lindsey Reymore.

**Data curation:** Niels Chr. Hansen.

**Formal analysis:** Niels Chr. Hansen, Lindsey Reymore.

**Investigation:** Lindsey Reymore.

**Methodology:** Niels Chr. Hansen, Lindsey Reymore.

**Project administration:** Niels Chr. Hansen, Lindsey Reymore.

**Software:** Niels Chr. Hansen.

**Visualization:** Niels Chr. Hansen, Lindsey Reymore.

**Writing – original draft:** Niels Chr. Hansen, Lindsey Reymore.

**Writing – review & editing:** Niels Chr. Hansen, Lindsey Reymore.

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
