## [Decision Letter · Decision Letter 0]

12 Mar 2024

PONE-D-23-35712Timbral cues underlie instrument-specific absolute pitch in expert oboistsPLOS ONE

Dear Dr. Hansen,

Thank you for submitting your manuscript to PLOS ONE. After careful consideration, we feel that it has merit but does not fully meet PLOS ONE’s publication criteria as it currently stands. Therefore, we invite you to submit a revised version of the manuscript that addresses the points raised during the review process.

We look forward to receiving your revised manuscript.

Kind regards,

Junchen Shang

Academic Editor

PLOS ONE

2. Please remove your figures from within your manuscript file, leaving only the individual TIFF/EPS image files, uploaded separately. These will be automatically included in the reviewers’ PDF.

4. In your cover letter, please confirm that the research you have described in your manuscript, including participant recruitment, data collection, modification, or processing, has not started and will not start until after your paper has been accepted to the journal (assuming data need to be collected or participants recruited specifically for your study). In order to proceed with your submission, you must provide confirmation.

Reviewers' comments:

Reviewer's Responses to Questions

**Comments to the Author**

1. Does the manuscript adhere to the experimental procedures and analyses described in the Registered Report Protocol?

If the manuscript reports any deviations from the planned experimental procedures and analyses, those must be reasonable and adequately justified.

Reviewer #1: Yes

Reviewer #2: Yes

2. If the manuscript reports exploratory analyses or experimental procedures not outlined in the original Registered Report Protocol, are these reasonable, justified and methodologically sound?

A Registered Report may include valid exploratory analyses not previously outlined in the Registered Report Protocol, as long as they are described as such.

Reviewer #1: Partly

Reviewer #2: Yes

3. Are the conclusions supported by the data and do they address the research question presented in the Registered Report Protocol?

The manuscript must describe a technically sound piece of scientific research with data that supports the conclusions. The conclusions must be drawn appropriately based on the research question(s) outlined in the Registered Report Protocol and on the data presented.

Reviewer #1: Partly

Reviewer #2: No

4. Have the authors made all data underlying the findings in their manuscript fully available?

Reviewer #1: Yes

Reviewer #2: Yes

5. Is the manuscript presented in an intelligible fashion and written in standard English?

Reviewer #1: Yes

Reviewer #2: Yes

6. Review Comments to the Author

Please use the space provided to explain your answers to the questions above. (Please upload your review as an attachment if it exceeds 20,000 characters)

Reviewer #1: I’ll divide my comments into items that I feel need to be fixed before publication to ensure methodological rigor, and those that are just recommendations which I feel could make the paper stronger but aren’t strictly necessary. Since this is a Registered Report (kudos to the authors for using this format), those comments in the former category mainly pertain to the exploratory analyses and discussion. As for those comments in the latter category, I recommend the editor leave it up to the authors’ discretion whether recommendations be implemented.

Main criticisms:

1. In Experiment 1, the difference between accuracy on oboe and piano trials is compared for each participant at the nominal significance level of 0.05. One might be concerned about a multiple comparisons problem here, since the probability of finding at least one false positive ISAP subject is greater than 0.721. A quick back-of-the-napkin calculation, assuming that the number of false positives among the a priori maximum n = 50 subjects is drawn from the distribution k ~ Binomial(50, 0.05), indicates that P(k >= 4) = 0.104 but P(k >= 5) = 0.038, so the present results (i.e. that k >= 10) do, in fact, allow you to reject the null hypothesis that there are no ISAP subjects in the population. This is probably worth mentioning in the manuscript, since the statistical inference the authors want to make does technically require such a correction.

2. The details of the exploratory analysis described under “Modeling Transposition as a Continuous Variable” are not totally clear in the manuscript. Looking at the R code, however, it seems that while a random intercept for each participant was included in the preregistered analysis (as it should be), it was dropped from this exploratory analysis. Please add the random intercept, or regression parameter estimates may be biased. (This should just be a one-line change to your code, and who knows, you might find something interesting!)

3. I’m not sure Figure 4 provides strong evidence for a continuum in ISAP (as opposed to just a continuum in AP ability) as the authors suggest in Discussion. While pitch-labeling accuracies for oboe notes do show quite a range in the present sample, they’re also very well correlated with accuracy for piano notes, which presumably reflect gradations of global AP ability. Consequently, much of the variation in ISAP ability might be explained by known variation in AP ability to which performance gains due to instrument-specific timbre cues are “added on.” It’s hard to say just by looking at it, but the drop in performance when timbre cues are disrupted in Experiment 2 seems to actually have very little variance across subjects (i.e. the lines denoting paired differences in Figure 7 mostly have similar slopes). I think that whether the (unique) contribution of instrument-specific cues to pitch-labeling ability is a stable individual difference makes an interesting scientific question for future research, but the present data don’t seem to indicate whether or not this is the case.

Other recommendations:

4. In the same vein as my comment re: multiple comparisons in Experiment 1 above, ignoring the possibility of Type I & II errors at the subject level will result in a biased estimate of population prevalence. Fortunately, this is relatively simple to remedy if you’d like. If you either establish some minimum effect size of interest for the subject-level chi-squared test (i.e. for oboe vs. piano accuracy) and then compute power for that effect size or just assume a power of 1 for a conservative lower bound, then population prevalence can be estimated directly from assumed power and the subject-level p-values using a simple Bayesian calculation (see Ince et al., https://doi.org/10.7554/eLife.62461). R functions for computing these population prevalence estimates, as well as differences in prevalence between conditions (e.g. in AP ability between piano and oboe conditions), with helpful example scripts, can be found here (https://github.com/robince/bayesian-prevalence). This is not a necessary addition, but could strengthen the paper; just keep in mind that “population prevalence” refers to the population sampled, rather than the general population.

5. As noted above, I do not think the current data provide specific evidence for a continuously grated distribution of subject-level effect sizes for instrument-specific timbre cue contributions to pitch-labeling ability as the authors suggest. It is worth noting, however, that previous research arguing that AP ability is continuously distributed (e.g. https://doi.org/10.1371/journal.pone.0244308) has been criticized for using complex rather than sine tones, precisely because of the concern that the use of timbre cues could confound “true” – or in the terminology of the present authors, “global” – pitch-labeling ability. (Full disclosure: I am an author on the linked paper, though it was already cited by the present authors.) The results of Experiment 2 seem to indicate that removing/disrupting timbre-cues reduces but does not destroy pitch-labeling accuracy, leaving substantial between-subject variance remaining. This study is a nice contribution to the literature in that regard, as it seems to falsify the above criticism.

6. It would be helpful to visualize the rejection threshold for labeling someone as ISAP (as well as lines for at-chance and for the threshold that defines above-chance performance) in Figure 4, rather than just coloring the points.

7. It is interesting that pitch-labeling accuracy, but not precision (were we to accept the null), is affected by the pitch shift, seeing as Fig. 6 indicates responses are non-random on incorrect trials generally. I’d think that, for example, if you pitch-shifted an A note to become a B with the timbre of an oboe-A, the timbre cues would bias the subject to respond something closer to A (or just “false alarm” respond A more often) than they would otherwise. Does the direction of the (not-absolute) semitone error depend on the direction of the pitch-shift? Or are subjects more likely to respond with the note with the original timbre? I think this would be interesting to add, but of course not necessary.

8. I would love to see more discussion of how it is mechanistically possible for accuracy to be reduced without a corresponding increase in mean absolute semitone deviation. I'm guessing something interesting may be going on here along the lines of my above comment, but maybe the authors have other ideas.

P.S. to the authors: I recommended "major revisions" since some of these changes could be substantive (depending on which suggestions you choose to implement or whether adding a random intercept changes results). However, I mentioned in my private note to the editor that revisions per my comments could end up being pretty minor. In my opinion, this manuscript is near the finish line.

Reviewer #2: The study defines “instrument-specific absolute pitch” (ISAP) as a “gain in absolute pitch identification ability for one’s own instrument type,” and the possessors of a “global absolute pitch” (AP) as those who can identify pitches “with high accuracy across a range of 66 timbres.” Bachem’s (1955) seminal work is missing here, since he had already pointed out this phenomenon he named “quasi” AP. Between the lines 87-91, the group identified as AP possessors are those who performed above 85% in identifying correct pitch stimuli. The references provided have measured the AP construct in different ways (i.e., duration of stimuli, percentage of correct responses, timbres), although two of them share commonalities (i.e., only correct pitches are considered for AP). Later, between lines 113-115, another work (Li [20]) is cited. According to Li’s work, “73% correct without semitone errors and 80% with semitone errors” are considered for AP’s ability. Reference [10] provides AP assignment that includes semitone errors. Therefore, AP phenomenon continue to be arbitrarily defined, and it is necessary to clarify why AP possessors in this work are considered those whose correct responses are above 85%. Why wouldn’t it be arbitrary if there is no consensus in the literature? The issue should be further explored and justified. In addition, if there is no consensus about the underlying factor for AP, it is even harder to affirm that ISAP is an existent cognitive ability, and the fact the participants were not sure about they possess it or not might be due to the lack of clarity of the construct model, even if they affirm that they are aware of its existence. At lines 214-216, on what grounds are the authors relying to support the claim that the accuracy is above chance level? At line 221, describe how recruitment was done. At lines 252 – 256, it seems that the authors are not considering the high level of ear training experience the participants have. In other words, the possibility that they made use of the relative pitch memory is not irrelevant (what was indeed declared at the self-report questionnaire by 28 participants out of 40 [line 440] after the tests). This is especially true considering that the participants could listen to the stimuli as many times as they want to. Between lines 259-264, reference [12] does not provide a validated tested model for ISAP. At Figure 5, the differences between the two groups in performance in both white and black keys are very slight: how is the grouping justified here so as not to appear arbitrary? At lines 406-406, pitch confusion is discussed. In this case, it is very possible that relative pitch is being used by participants. This possibility should be considered here. Line 486 mentions the Registered Report to refer to the simulations to determine the minimum sample size. I suggest it is briefly provided in this paper. The same happens in lines 441-443: I suggest that the possible interferences of the strategies referred to in another article are summarized here. At the Conclusions, at line 815, I would recommend the use of “The current study suggests that etc.”, instead of “The current study provides the first evidence etc.” In the same way, at line 664-665, I would recommend: “our data suggest,” instead of “our data confirm.” Also, at line 666, the reference 47 refers to a study supposedly conducted with non-musicians according to the text, but only musicians were tested there. The findings and the discussions about pitch shifting described at p. 14-15 are quite interesting, and it offers important clues to potential strategies used by instrumentalists in pitch perception, but maybe not particularly to AP. At the final references, n. 42 and 43 should be merged.

7. PLOS authors have the option to publish the peer review history of their article (what does this mean?). If published, this will include your full peer review and any attached files.

Reviewer #1: **Yes: **John P. Veillette

Reviewer #2: No

---

## [Author Response · Author response to Decision Letter 0]

27 May 2024

Please see the attached "Response to reviewers" file.

---

## [Decision Letter · Decision Letter 1]

26 Jun 2024

Timbral cues underlie instrument-specific absolute pitch in expert oboists

PONE-D-23-35712R1

Dear Dr. Hansen,

We’re pleased to inform you that your manuscript has been judged scientifically suitable for publication and will be formally accepted for publication once it meets all outstanding technical requirements.

Kind regards,

Junchen Shang

Academic Editor

PLOS ONE

Additional Editor Comments (optional):

Reviewers' comments:

Reviewer's Responses to Questions

**Comments to the Author**

1. Does the manuscript adhere to the experimental procedures and analyses described in the Registered Report Protocol?

If the manuscript reports any deviations from the planned experimental procedures and analyses, those must be reasonable and adequately justified.

Reviewer #1: Yes

Reviewer #2: Yes

2. If the manuscript reports exploratory analyses or experimental procedures not outlined in the original Registered Report Protocol, are these reasonable, justified and methodologically sound?

A Registered Report may include valid exploratory analyses not previously outlined in the Registered Report Protocol, as long as they are described as such.

Reviewer #1: Yes

Reviewer #2: Yes

3. Are the conclusions supported by the data and do they address the research question presented in the Registered Report Protocol?

The manuscript must describe a technically sound piece of scientific research with data that supports the conclusions. The conclusions must be drawn appropriately based on the research question(s) outlined in the Registered Report Protocol and on the data presented.

Reviewer #1: Yes

Reviewer #2: Yes

4. Have the authors made all data underlying the findings in their manuscript fully available?

Reviewer #1: Yes

Reviewer #2: Yes

5. Is the manuscript presented in an intelligible fashion and written in standard English?

Reviewer #1: Yes

Reviewer #2: Yes

6. Review Comments to the Author

Please use the space provided to explain your answers to the questions above. (Please upload your review as an attachment if it exceeds 20,000 characters)

Reviewer #1: The authors have done a good job addressing my comments. The only comments they did not chose to address in the revised manuscript -- though they kindly humored me in their point-by-point responses -- were those that I had specifically stated were just suggestions (rather than concerns), which is very appropriate especially for a Registered Report.

One note: I am glad the authors chose to use prevalence statistics to quantify the prevalence of ISAP in the sampled population. I suggest, however, that they add the words "in the sampled population" to their reporting of results, as the current phrasing could be confused with a statement about ISAP prevalence in the general population. (I know this is a bit of a silly concern, since the reported prevalence would be ridiculously high for a population that includes non-musicians, but it is worth being clear, I think).

Reviewer #2: The questions raised in the previous review were clarified and answered in their entirety, whether in the changes and justifications provided in the manuscript or in the form of clarifications directly in response to the reviewer, who is grateful for the consideration.

7. PLOS authors have the option to publish the peer review history of their article (what does this mean?). If published, this will include your full peer review and any attached files.

Reviewer #1: **Yes: **John P. Veillette

Reviewer #2: No

---

## [Editor Report · Acceptance letter]

12 Aug 2024

PONE-D-23-35712R1 

PLOS ONE

Dear Dr. Hansen, 

I'm pleased to inform you that your manuscript has been deemed suitable for publication in PLOS ONE. Congratulations! Your manuscript is now being handed over to our production team.

Kind regards, 

on behalf of

Dr. Junchen Shang 

Academic Editor

PLOS ONE